# JAK/STAT as a Potential Therapeutic Target for Osteolytic Diseases

**DOI:** 10.3390/ijms241210290

**Published:** 2023-06-17

**Authors:** Mariely A. Godoi, Angelo C. Camilli, Karen G. A. Gonzales, Vitória B. Costa, Evangelos Papathanasiou, Fábio R. M. Leite, Morgana R. Guimarães-Stabili

**Affiliations:** 1Department of Diagnosis and Surgery, School of Dentistry at Araraquara, UNESP, Araraquara 14801-385, Brazil; mariely.a.godoi@unesp.br (M.A.G.);; 2Department of Periodontology, Tufts University School of Dental Medicine, Boston, MA 02111, USA; evangelos.papathanasiou@tufts.edu; 3National Dental Research Institute Singapore, National Dental Centre, Singapore 168938, Singapore; fabio@duke-nus.edu.sg; 4Oral Health Academic Clinical Programme, Duke-NUS Medical School, Singapore 169857, Singapore

**Keywords:** JAK/STAT inhibitor, osteoclasts, osteoblasts, cytokines, JAK/STAT signaling pathway, osteolysis, osteolytic disease

## Abstract

Several cytokines with major biological functions in inflammatory diseases exert their functions through the Janus kinase (JAK)-signal transducer and activator of transcription (STAT) signal transduction pathway. JAKs phosphorylate the cytoplasmic domain of the receptor, inducing the activation of its substrates, mainly the proteins known as STATs. STATs bind to these phosphorylated tyrosine residues and translocate from the cytoplasm to the nucleus, further regulating the transcription of several genes that regulate the inflammatory response. The JAK/STAT signaling pathway plays a critical role in the pathogenesis of inflammatory diseases. There is also increasing evidence indicating that the persistent activation of the JAK/STAT signaling pathway is related to several inflammatory bone (osteolytic) diseases. However, the specific mechanism remains to be clarified. JAK/STAT signaling pathway inhibitors have gained major scientific interest to explore their potential in the prevention of the destruction of mineralized tissues in osteolytic diseases. Here, our review highlights the importance of the JAK/STAT signaling pathway in inflammation-induced bone resorption and presents the results of clinical studies and experimental models of JAK inhibitors in osteolytic diseases.

## 1. Introduction

Bone architecture is one of the most important systems in the human body, supporting tissues and protecting vital organs [1]. In addition, it is in the bone tissue that a large part of the reserves of calcium and phosphate are found, minerals of extreme importance to maintaining systemic health [1].

Bone constantly undergoes a process of resorption of older bones and the formation of new ones [1]. This is characterized as bone remodeling, resulting from a balanced interplay between two important cells, osteoclasts and osteoblasts [2]. The functions of these cells are at opposite ends; while osteoblasts are responsible for forming bone tissue, osteoclasts resorb it [2]. Usually, these processes are well orchestrated, so there is no discrepancy in bone mass maintaining homeostasis [2]. However, an increase or decrease in the number and/or activity of these cells is related to the development of diseases such as inflammatory arthritis, myelofibrosis, periodontitis, osteoporosis, osteopetrosis, and bacterial-induced osteolysis, among others [1,2].

Osteoblast and osteoclast activities are regulated by numerous cytokines expressed in the bone microenvironment [3]. At the cellular level, the information transmitted by these cytokines triggers a cascade of signaling pathways that process information [3]. Among them, the Janus tyrosine kinase (JAK) and signal transducers and activators of transcription-mediated signaling (STAT) are responsible for the signal transduction of more than fifty cytokines, growth factors, and hormones with pivotal roles in bone homeostasis [3].

Therefore, this article reviews the role of the JAK/STAT signaling pathway during bone remodeling affected by inflammation and how its inhibition can be used as a therapeutic target in osteolytic disorders. To the best of our knowledge, this is the first review to gather results from clinical and preclinical studies on the use of JAK inhibitors in the treatment of inflammatory bone diseases.

## 2. JAK/STAT Cell Signaling Pathway

The JAK-STAT pathway was originally described as primarily activated in response to interferon (IFN)-gamma and members of the interleukin-6 (IL-6) family [3]. It is now well understood that it serves as a mediator for numerous cytokines, hormones, and growth factors, suggesting that it plays a role in bone development, metabolism, and healing [4].

### 2.1. Family Members

The JAK family includes four tyrosine kinases (JAK1, JAK2, JAK3, and TYK2) that are selectively associated with different chains of cytokine receptors for the phosphorylation of multiple protein-derived residues [5]. The STATs, in turn, bind to these phosphorylated tyrosine residues, dimerize, and translocate from the cytoplasm to the nucleus, where they bind to DNA and can then activate or block the signal transcription [6]. There are seven types of STATs in mammals, STAT1, STAT2, STAT3, STAT4, STAT5A, STAT5B, and STAT6, and although they are activated by overlapping subsets of cytokines, different STATs may have non-redundant biological roles [6].

### 2.2. JAK/STAT Signaling Pathway Regulatory Mechanisms

Cell activation through the JAK/STAT signaling pathway is very fast and effective after cytokines’ binding to their respective receptors [7]. JAKs become active after the multimerization of the receptor mediated by ligands; in this way, there is a phosphorylation of the main substrate, STAT [7] (Figure 1).

A family of critical proteins in the transduction of signals and in the modulation of the JAK/STAT pathway are the suppressors of cytokine signaling (SOCS) proteins [8]. The SOCS family consists of eight proteins activated by several pro- and anti-inflammatory cytokines, and it interferes with JAK/STAT signaling through negative feedback [8]. Excessive JAK/STAT stimulation by cytokines such as IL-1, IL-6, IL-15, TNF-α, and IFN has been associated with rheumatoid arthritis (RA), multiple sclerosis, inflammatory bowel disease, and periodontal disease, to name a few [8].

Hence, JAK/STAT signaling is crucial in initiating the innate immune response and orchestrating the mechanisms responsible for adaptive immunity [9]. Therefore, inhibiting its activation restricts undesirable responses of inflammatory processes in various pathological conditions, many of which are of osteolytic origin [5,9,10].

## 3. JAK/STAT in Bone Metabolism

Several studies have shown that the JAK/STAT signaling pathway is closely related to diseases of autoimmune origin, probably through the intertwining of the immune and bone systems [11,12,13,14]. Although each bone cell has its own specific function, osteoblasts directly influence the differentiation of osteoclasts through the production of several molecules, e.g., the receptor activator of the nuclear factor κB ligand (RANKL) and osteoprotegerin (OPG) [11]. Physiological bone remodeling occurs through tight regulation of the RANK/RANKL/OPG triad [12]. RANKL is pivotal during bone resorption as it binds to RANK on the osteoclast precursors’ surface and induces differentiation into mature osteoclasts [13]. However, this interaction is negatively regulated in the presence of OPG, as this protein has an affinity with RANKL, serving as a decoy and impeding RANKL binding to RANK [12,13,14].

Some studies have shown that the JAK2/STAT5B pathway plays an important role in the mechanisms of growth hormone activation, a regulator of bone growth and metabolism [15,16,17]. This pathway is also fundamental in the formation of osteoblasts since it acts on important transcription factors involved in osteoblastogenesis, such as Runt-related transcription factor 2 (RUNX-2), bone morphogenetic protein-7 (BMP-7), and T- box transcription factor 3 (TBX-3) [12,18,19,20]. Studies evaluating the effect of JAK on osteoblast differentiation indicated an increase in the activities of these cells after treatment with a JAK inhibitor, with up-regulation of important markers for osteoblast function such as osteocalcin and Wnt signaling [21]. Similarly, JAK inhibition dose-dependently enhanced osteogenic differentiation of human mesenchymal stromal cells (hMSCs), as demonstrated by increased calcium deposition by cells and increased transcript (mRNA) levels of RUNX-2 and collagen [22]. Expression of other osteoblast-specific markers, including alkaline phosphatase (ALP), type I collagen α1 (Col1a1), OCN, and OPN, were also reduced after JAK activation in vitro [23]. In vivo, the regulatory role of JAK on bone formation has also been demonstrated in mice with bone loss induced by estrogen deficiency (ovariectomy) or inflammation (arthritis). In both models, JAK inhibition increased bone mass, which was consistent with reducing the ratio of receptor activator of RANKL/osteoprotegerin in serum [21]. The effect of JAK inhibition on osteogenesis in vitro and in vivo, however, may be dependent on the type of inflammatory stimulus and/or the specificity of the inhibitor used (i.e., the methodology used in each study), since, contrary to the results described above, JAK2 inhibition decreased the differentiation and osteogenic activity of BMSC stimulated with CoCl_2_, a cell hypoxia stimulant [24]. Protein levels of COL1α1, ALP, RUNX2, and OSX were inhibited in cells after treatment with JAK inhibitors, as was the formation of mineralized nodules. In parallel, the administration of a JAK2 inhibitor in mice strongly inhibited bone regeneration and healing in bone defects made in the femur of mice [24].

In addition to the role of JAK in osteoblasts, research has demonstrated the crucial role of JAK in osteoclasts and, consequently, in bone resorption [21,22]. hMSCs stimulated with RANKL and M-CSF and treated with a JAK inhibitor demonstrated reduced osteoclast differentiation and activity, and reduced expression of osteoclast markers (cathepsin K and RANK) [22]. In vivo, systemic administration of JAK inhibitor (60 mg/kg) orally suppressed bone resorption induced by LPS in mouse calvaria by blocking the function of mature osteoclasts and preventing the migration of osteoclast precursors to the bone surface [25]. There are several possible mechanisms for this inhibitory effect, one of them is that JAK inhibitors may suppress inflammation-related IL-6 signaling. Precursors of murine osteoclasts (RAW 264.7) stimulated with IL-6 and treated with JAK inhibitors demonstrated inhibition of IL6-induced STAT3 phosphorylation [25]. Studies by our research group evaluating the effect of JAK inhibition on alveolar bone resorption in a model of periodontitis in rats also observed a protective effect of the inhibitor on bone loss and a reduction of IL-6 levels in periodontal tissues, also suggesting the involvement of IL-6 signaling in the effects of JAK on inflammatory bone resorption [26]. The protective effect of JAK inhibition on bone resorption was also observed in ovariectomized mice [27]. Intraperitoneal administration of a JAK1/2 inhibitor reduced ovariectomy-induced bone loss and decreased the number of osteoclasts as well as the expression of CTSK-positively labeled cells [27].

In addition to the importance of JAK in bone cells, it is worth highlighting the role of cytokines that are also involved during the inflammatory process and their relationship with the JAK/STAT signaling pathway [28]. Studies report that with the activation of STAT3 and the receptor subunit gp130, IL-6 can inhibit the differentiation of osteoclasts by RANKL [28]; however, it stimulates the production of IL-1 [29] and RANKL in osteoblasts, inducing greater bone resorption [30,31].

Through STAT6 phosphorylation, there is an inhibition of osteoclast activity by IL-4, and conversely, through phosphorylation of STAT5, IL-7 induces the formation of osteoclasts [32]. IL-3, on the other hand, stimulates the process of osteoclastogenesis probably through the JAK2/STAT5 pathway [32,33,34] and IL-17A through the JAK2/STAT3 pathway [35,36,37].

Further to the role of JAK on bone turnover under inflammatory conditions, it is important to highlight the relevance of the pathway for bone metabolism in physiological contexts. Genetically JAK1-deficient mice present less bone growth and lower body mass, fail to nurse despite the normal nurturing behavior of their mothers, and die perinatally [38]. Corroborating this result, in another study, JAK1 knockout mice died during the perinatal period, showing a significantly smaller skeletal structure when compared to counterparts without the deficiency [38].

Finally, it is important to emphasize that not all JAKs and STATs are equally potent regulators of bone biology, which means that tuning the level of bone remodeling may be possible by acting on specific JAKs and/or STATs [1]. Further studies shall explore their modulation or inhibition capacities as therapeutic targets in inflammatory disorders of osteolytic origin.

## 4. Inhibition of the JAK/STAT Pathway as a Therapeutic Target

Drugs that inhibit the JAK/STAT pathway to reduce signs and symptoms of inflammatory diseases have gained great interest. Some are already marketed and are increasingly subjected to research on their effects at different target sites [10]. The delay in the ample use and commercialization of JAK inhibitors is widely related to purchase costs, which differ broadly among countries and even internally due to fluctuations caused by seasonal demands [39]. For example, the price is fixed in France, but it is based on free demand in the United States, JAK inhibitors are generally more expensive than conventional synthetic disease-modifying antirheumatic drugs (DMARDs—methotrexate, sulfasalazine, leflunomide, and hydroxychloroquine) but are comparable in price to biologic DMARDs (infliximab, adalimumab, etanercept, abatacept, and tocilizumab) [39].

The JAK inhibitors that are currently approved by the Food and Drug Administration(FDA) and the European Medicines Agency (EMA) are Ruxolitinib (Jakavi^®^), which aims to inhibit JAK1 and JAK2 in patients with myelofibrosis and polycythemia vera; Tofacitinib (Xeljanz, Jakvinus, CP-690550), which inhibits JAK1 and JAK3 in rheumatoid and psoriatic arthritis; and finally, Baricitinib, targeting JAK1 and JAK2, for the treatment of rheumatoid arthritis [39]. Other JAK inhibitors are currently the subject of numerous studies, such as Pefacitinib, Filgotinib, Upacitinib, Itacitinib, Momelitinib, Gandotinib, Lestaurtinib, Decernotinib, Filgotinib, and Pacritinib, and showed promising results in Phase III clinical trials in rheumatoid arthritis, atopic dermatitis disease, Crohn’s disease, or myelofibrosis [39,40,41,42].

There are adverse effects reported for JAK inhibitors, mainly associated with Tofacitinib, e.g., malignancy, non-viral opportunistic infections, gastrointestinal perforation, and herpes zoster [40]. However, a greater use of these inhibitors over a longer exposure time will provide more evidence about the risks associated with these compounds [40,43,44,45,46].

We will address the results of inhibitors of the JAK/STAT signaling pathway reported in the literature in bone diseases that affect a large part of the world’s population.

## 5. JAK/STAT Inhibitors in Pathological Bone Diseases

### 5.1. Rheumatoid Arthritis

One of the most prevalent bone diseases is rheumatoid arthritis, a chronic systemic inflammatory condition characterized by joint destruction that progressively affects the synovial lining of the joints, causing pain, deformity, and loss of function in the affected limbs [47]. Conventional (such as methotrexate and sulfasalazine) and biological (TNF and IL-6 inhibitors) DMARDs have been the mainstay for treating patients with rheumatoid arthritis [48]. However, due to the lack of efficacy and side effects, new drugs were explored to find more viable alternative treatments for the disease [47,48].

A JAK3 inhibitor was approved by the FDA as a therapeutic strategy for rheumatoid arthritis as an alternative for patients who do not respond well to other drugs [49]. The inhibitor tofacitinib has been shown to decrease the production of IL-17 and IFN-γ and the proliferation of CD4+ T cells in cells derived from patients with arthritis [50] (Table 1). A systematic review and meta-analysis evaluating the results of more than 4000 patients using JAK3 inhibitors for the treatment of rheumatoid arthritis concluded that the treated group showed significant improvement in signs and symptoms compared to other groups [51]. Moreover, the inhibitor presented a safe pharmacological profile and prolonged efficacy for up to 8 years of follow-up [51]. Side effects related to immunosuppression have been reported by some patients and are related to the development of opportunistic infections, such as herpes zoster and respiratory infections; however, the risk for the development of these complications is the same as presented by the administration of other DMARDs [52]. Preliminary studies in rats also suggest an increase in bone mass, compatible with a reduction in the RANKL/OPG ratio in serum, as well as an increase in osteoblast function after the administration of tofacitinib and baricitinib [21] (Table 2).

In a randomized, double-blind, placebo-controlled, parallel-group clinical trial, final scores of patients with rheumatoid arthritis improved, as did greater inhibition of radiographic progression of bone loss with oral ingestion of tofacitinib (10 mg and 5 mg, twice a day, daily, for 3 months) [49,53]. Consistently, another study in patients with inadequate response or intolerance to conventional synthetic or biologic DMARDs demonstrated a significant improvement in rheumatoid arthritis signs and symptoms after 24 weeks with the use of baricitinib prescribed according to local recommendations for biologic and targeted synthetic therapy aligned with European League Against Rheumatism (EULAR) guidelines (<7.5 mg/day) [54].

By evaluating the use of the JAK inhibitor Tofacitinib in a rat adjuvant-induced arthritis (AIA) model, LaBranche et al. obtained satisfactory results, such as reduction of edema, inflammation, bone resorption, and plasmatic RANKL and IL-6 levels [50]. In vitro results corroborated the decrease in RANKL levels by T lymphocytes after treatment with a JAK inhibitor [55].

In rats with adjuvant-induced arthritis, tofacitinib markedly reduced the clinical status of treated rats compared to the control group [56]. The main findings were a reduction in joint inflammation and a decrease in serum C-reactive protein levels, which were reflected in a significant reduction in mean paw diameter and an increase in body weight. Furthermore, tofacitinib significantly reduced the frequency of Clusters of Differentiation 4 (CD4)+, IFN-γ+, and T cells and the levels of IL-1β mRNA expression in the spleen of treated rats [56].

Some recent studies provide valuable information about the efficacy and safety of different treatment options for rheumatoid arthritis and their impact on patient-reported outcomes (PROs) and disease progression [57,58,59,60,61]. Research has focused on identifying distinct trajectories of disease activity in methotrexate-naïve patients with rheumatoid arthritis receiving tofacitinib for 24 months [57]. The results revealed that treatment with tofacitinib (5 mg twice daily) led to significant improvements in disease activity, with the majority of patients experiencing sustained low activity or disease remission. However, a small proportion of patients exhibited trajectories of high disease activity, suggesting the need for individualized treatment approaches [57]. In contrast, two studies examined the impact of initial therapy with upadacitinib (15 mg/day) or adalimumab on achieving treatment goals in patients with RA. Both studies demonstrated that treatment with upadacitinib led to significantly higher rates of achieving clinical remission or low disease activity compared with adalimumab [58,60]. Still, based on the results of another study that focused on PROs in Asian patients with RA, treatment with peficitinib (100 and 150 mg/day) demonstrated significant improvements in pain, physical function, and other PROs compared to placebo [59]. However, a post hoc analysis of a Japanese phase 3 study of peficitinib (100 mg/day) and methotrexate showed positive results in reducing radiographic progression, even though some patients did not respond to treatment, indicating the need for further investigations into optimal treatment strategies [61].

Overall, these studies highlight the importance of individualized treatment approaches for RA and the need to consider clinical and PRO factors when assessing treatment efficacy. Furthermore, the results suggest that newer treatments such as upadacitinib and tofacitinib may provide superior efficacy compared to traditional treatments such as methotrexate and adalimumab.

A better understanding of JAK/STAT pathway activation for the occurrence and development of rheumatoid arthritis will allow the deepening of strategies and therapeutic targets and the development of JAK inhibitor antirheumatic drugs.

### 5.2. Myelofibrosis

Myelofibrosis is characterized by an overproduction of myeloid stem cells, bone marrow fibrosis, cytopenia, and extramedullary hematopoiesis, which exacerbates the release of inflammatory cytokines [62]. Bone involvement in primary myelofibrosis has many forms; it affects bone marrow, leading to bone marrow fibrosis, and it can cause periostitis in addition to bone and joint pain. The only curative treatment for myelofibrosis is allogeneic hematopoietic stem cell transplantation. Still, because of life-threatening risks, especially for elderly patients and those with comorbidities, new therapeutic targets have been studied as alternative treatments for the disease [62].

A promising example is AZD1480, which blocks cell proliferation at low micromolar concentrations and induces apoptosis in myeloma cell lines via concomitant inhibition of the phosphorylation of signaling proteins JAK2, STAT3, and MAPK [63]. In clinical trials, an improvement in the symptoms of patients using ruxolitinib was observed due to the reduction in the degree of bone marrow fibrosis and normalization of bone lesions at concentrations of 20 mg for 16 weeks in one study and 15 mg for 12 months in another, both used twice a day, daily [64,65]. In another study, continued therapy with ruxolitinib (initially prescribed at a dose of 15 or 20 mg twice a day, depending on the baseline platelet count) was associated with marked and lasting reductions in splenomegaly and disease-related symptoms, improved quality of life, and modest toxic effects [65].

A problem associated with myelofibrosis is the appearance of infectious diseases [52]. A reduction in the cytokine-induced activity and function of natural killer cells in patients with myelofibrosis was observed [52], suggesting a potential preventive role of ruxolitinib in secondary infections.

Some recent studies have investigated different treatments for myelofibrosis, each targeting different disease pathways [66,67,68]. The first study compared the effectiveness of momelotinib (100 and 200 mg/day) and danazol in relieving anemia and other symptoms in patients with myelofibrosis. The study found that both drugs improved symptoms, but momelotinib was associated with fewer adverse events and better splenic response rates than danazol [66]. On the other hand, the study of fedratinib (300, 400, and 500 mg/day) found that the drug was effective in treating myelofibrosis in patients with low platelet counts, a common complication of the disease. The drug was generally well tolerated, with few serious adverse events reported [67]. Finally, a study on jaktinib (100 and 200 mg/day) evaluated the safety and efficacy of the drug in Janus kinase inhibitor-naïve patients with myelofibrosis. The drug was well tolerated and effective in reducing spleen size and improving symptoms, with no unexpected safety concerns [68].

Overall, these studies highlight the importance of targeted therapies in the treatment of myelofibrosis and suggest that different drugs may be appropriate for different patient populations, depending on individual disease characteristics.

Further research using the JAK/STAT signaling pathway inhibitors should be carried out to define a protocol for clinical application in myelofibrosis.

**Table 1 ijms-24-10290-t001:** Summarization of the findings from JAK-STAT inhibitors in in vivo and in vitro studies.

Drugs	Disease	Outcomes
**Inhibitor AZD1480** [63]	Myelofibrosis	**in vitro:** - Blocking of cell proliferation and induction of apoptosis of myeloma cell lines; - Cell death of KMS-11 cells grown in the presence of HS-5 bone marrow-derived stromal cells - Inhibition of tumor growth in a KMS-11 xenograft mouse model, accompanied with inhibition of phospho-FGFR3, phospho-JAK2, phospho-STAT3 and Cyclin D2 levels;
**Tofacitinib** [50]	Rheumatoid Arthritis	**in vitro**: - Inhibition of the production of IL-17 and IFN in a dose-dependent manner; - Effect in proliferation and transcription;
**Baricitinib** [55]	Rheumatoid Arthritis	**in vivo**: - Reduction of edema; - Reduction of inflammation and bone resorption; - Reduction of RANKL and IL-6 levels; **in vitro**: - Did not affect osteoclast function and activity; - Decreased RANKL levels produced by T lymphocytes in a dose-dependent manner
**Tofacitinib** [56]	Rheumatoid Arthritis	**in vivo**: - Reduction of the clinical status of treated rats in comparison to the control group. - Reduction of joint inflammation and down-regulated serum CRP levels reflected the clinical manifestations of the treated rats. - Tofacitinib down-regulated significantly the frequency of CD4+IFN-γ+ T cells and reduced IL-1β mRNA expression levels in the spleen of the treated rats;
**Tofacitinib** [21]	Rheumatoid Arthritis	**in vivo**: - Reduction in the severity of inflammation and in the physical score of arthritis; - Reduction in C-reactive protein levels; **in vitro**: - Reduction of splenic CD4 T cell levels;
**Ruxolitinib** [52]	Myelofibrosis	- Reduction in NK cell numbers; - Endogenous functional defects of NK cells in MPN were further aggravated; - Reduction in cytokine-induced NK cell activation; - Reduced killing activity of primary NK cells was associated with an impaired capacity to form lytic synapses with NK target cells; - Ruxolitinib impairs NK cell function in MPN patients;
**Baricitinib** [21]	Rheumatoid Arthritis	**in vivo**: - Increased bone mass, consistent with reducing the ratio of receptor activator of NF-ƙB ligand/osteoprotegerin in serum; **in vitro**: - Increased osteoblast function but no direct effects on osteoclasts;
**Tofacitinib and Baricitinib** [21]	Osteoporosis and Rheumatoid Arthritis	**in vivo**: - Increased bone mass, consistent with reducing the ratio of receptor activator of NF-ƙB ligand/osteoprotegerin in serum; **in vitro**: - Increased osteoblast function; - Robust up-regulation of markers for osteoblast function, such as osteocalcin and Wnt signaling;
**CYT387** [27]	Osteoporosis	**in vitro**: - Attenuates the formation of osteoclasts induced by RANKL; - Suppression of the bone reabsorption function of osteoclasts; - Repression of expression levels of osteoclast-specific genes; - Suppression of the expression and activation of NFATc1 induced by RANKL; - Inhibition of the MAPK signal activated by RANKL and intracellular Ca^2+^ influx; **in vivo**: - Prevention of bone loss in oophorectomized mouse models; - No effect of CYT387 on osteoblast differentiation;
**Stattic (STAT3 Inhibitor)** [56]	Osteoporosis	**in vitro**: - Inhibited osteoclastic differentiation and bone resorption in RANKL-induced RAW264.7 cells; - Suppressed RANKL-induced upregulation of tartrate-resistant acid phosphatase osteoclast-related genes in RAW264.7 cells; - Inhibition of RANKL-induced activation of STAT3 and NF-κB pathways, without significantly affecting MAPK signaling; - Restriction of osteoclastogenesis and bone loss by disrupting RANKL-induced STAT3 and NF-κB signaling;
**Ruxolitinib** [69]	Osteoporosis	**in vivo**: - Prevention of bone loss in ovariectomized mice; - Relieved senescence and enhanced osteogenic differentiation;
**miR-151a-3p** [70]	Osteoporosis	**in vitro**: - Inhibition of cell viability and promotion of lactate dehydrogenase release, increasing the RANKL/OPG ratio and decreasing Runx2 and BMP2 expressions; **in vivo**: - In an ovariectomized rat model, miR-151a-3p decreased bone mineral density and biomechanical parameters of femurs, targeting SOCS5; - miR-151a-3p contributes to the pathogenesis of postmenopausal osteoporosis and promotes its progress;

Abbreviations: IL: Interleukins; INF: Interferon; RANKL: Receptor Activator of Nuclear Factor-kappa Be ta Ligand; NF-ƙB: Factor Nuclear Kappa B: STAT: Signal Transducer and Activator of Transcription; JAK: Janus Kinase; NK cells: Natural Killers cells: MPN: Myeloproliferative Neoplasms; NFATc1: Nuclear Factor of Activated T Cells 1; MAPK: Mitogen-Activated Protein Kinase; STAT3: Signal Transducer and Activator of Transcription 3.

**Table 2 ijms-24-10290-t002:** Summarization of the findings from clinical studies on JAK-STAT inhibitors.

Drugs	Disease	Outcomes	Side Effects	Oral Protocol
**Ruxolitinib** [71]	Myelofibrosis	- Reduction of spleen volume; - Improved quality of life; - Improvement of the symptoms of the disease;	Diarrhea, peripheral edema, asthenia, dyspnea, nasopharyngitis, pyrexia, cough, nausea, arthralgia, fatigue, pain in extremities, abdominal pain, headache, back pain, pruritus.	15 and 20 mg twice daily
**Ruxolitinib** [64]	Myelofibrosis	- Improvement in symptoms and other signs of myeloproliferation;	None reported	20 mg twice daily
**Ruxolitinib** [65]	Myelofibrosis	- Reduction of bone marrow fibrosis grade and resolution of osteolytic lesions;	None reported	15 mg twice daily
**Momelotinib** [66]	Myelofibrosis	- Improvement in quality of life, reduction in the need for blood transfusions, and improvement in fatigue, pain, early satiety and weight loss;	- Anemia, thrombocytopenia, nausea, diarrhoea, headache, fatigue, arthralgia;	100 and 200 mg/day
**Fedratinib** [67]	Myelofibrosis	- Increased platelet count and reduction in splenomegaly;	- Diarrhoea, nausea, anemia and headache;	300, 400 and 500 mg/day
**Jaktinib** [68]	Myelofibrosis	- Improvement of fatigue, pain and gastrointestinal symptoms; - Inhibition of the activity of the JAK2 protein in the cells of patients with proliferation of myelofibrotic cells;	- Anemia, nausea and diarrhea;	100 and 200 mg/day
**Tofacitinib** [49,53]	Rheumatoid arthritis	- Improvement in: * American College of Rheumatology scale; * Erythrocyte sedimentation rate; * hsCRP levels; * Health Assessment Questionnaire-Disability Index; - greater inhibition of radiographic signs of disease progression	Report of viral infections (Herpes Zoster) and gastrointestinal disorders	5 and 10 twice daily
**Baricitinib** [54]	Rheumatoid arthritis	- All clinical parameters of rheumatoid arthritis decreased significantly (DAS28-CRP, SDAI, ESR, CRP, ACPA and FR);	None reported	Recommendation EULAR (<7.5 mg/day)
**Tofacitinib** [57]	Rheumatoid arthritis	- Reduction of joint swelling; - Reduction in C-reactive protein levels; - Improvement of reported quality of life;	None reported	5 mg twice daily
**Upadacitinib** [58]	Rheumatoid arthritis	- Improvement in disease activity scores, pain and functional disability	None reported	15 mg/day
**Peficitinib** [59]	Rheumatoid arthritis	- Significant improvements in Disease Activity Score (DAS), American Criteria Response (ACR), Patient Reported Outcomes (PROs) and Medical Reported Outcomes (MROs);	- Headache, nasopharyngitis, diarrhea, nausea and increased alanine aminotransferase (ALT) and aspartate aminotransferase (AST), but symptoms well tolerated.	100 and 150 mg/day
**Upadacitinib** [60]	Rheumatoid arthritis	- Greater improvement in joint pain, morning stiffness, fatigue and quality of life;	- Respiratory infections, nasopharyngitis and headache;	15 mg/day
**Peficitinib** [61]	Rheumatoid arthritis	- Reduction of radiographic progression;	- Upper respiratory tract infections, anaemia, decreased white blood cell count, increased liver enzymes, headache, pneumonia and shingles;	100 mg/day
**Tofacitinib** [72]	Periodontal disease	- Tofacitinib therapy reduced periodontal inflammation as indicated by the mean values of the gingival index, pocket depth, clinical attachment level, percentage of sites with bleeding on probing; - Serum levels of rheumatoid factor, matrix metalloproteinase-3, and IL-6 were decreased compared to the values at baseline;	None reported	10 mg/day
**Baracitinib** [54]	Periodontal disease	- Patients with chronic periodontitis showed a significant decrease in periodontal inflammation as suggested by improvement in the number of sites with bleeding on probing and pocket depth compared to the values of baseline;	None reported	Recommendation EULAR (<7.5 mg/day)

Abbreviations: hsCRP: High sensitivity C-reactive protein; DAS28-CRP: Disease Activity Score including 28 joints using C-Reactive Protein; SDAI: Simplified Disease Activity Index; ESR: Erythrocyte Sedimentation Rate; CRP: C-Reactive Protein; ACPA: Anti-Citrullinated Peptide Antibody; RF: Rheumatoid Factor; DAS: Disease Activity Index Scores, ACR: American Criteria Response, PROs: Patient Reported Outcomes and Medical Reported Outcomes (MROs); ALT: alanine aminotransferase; AST: aspartate aminotransferase; EULAR: European League Against Rheumatism.

### 5.3. Periodontitis

Periodontitis is one of the major causes of tooth loss in adults, affecting a high number of patients worldwide and creating a major public health concern that necessitates the discovery of novel therapies [73]. Periodontitis is a chronic inflammatory condition resulting from the actions of multiple causes that leads to microenvironmental changes in the tooth-supporting tissues and is exacerbated by a further dysbiosis of the microbial biofilm that deepens the inflammatory response [74]. The bacteria in this biofilm release numerous virulence factors, including lipopolysaccharides and antigens, which activate cytokines, chemokines, and transcription factors that may contribute to connective tissue destruction and bone resorption [75]. The standard treatment combines professional biofilm with a personalized at-home hygiene program. However, some patients with severe forms of the disease are refractory to treatment, and new host modulation therapies are necessary [76,77,78].

During periodontitis progression and chronification, numerous signaling pathways are activated depending on the nature of the causes [79]. Although few studies have investigated the role of JAK activation in periodontitis, STAT3 and STAT5 were shown to be activated in gingival tissues of periodontally diseased rats, indicating that the JAK/STAT pathway may play a relevant role in the pathogenesis of the disease [79].

Another study evaluated the effect of the absence of suppressor cytokine signaling protein 3 (SOCS-3) on inflammatory bone resorption in mice subjected to experimentally induced periodontitis. SOCS are a family of cell signaling molecules activated by microbial and immunological stimuli that downregulate cytokine signaling, inhibiting the JAK/STAT signal transduction pathway [80]. The results demonstrated that periodontally diseased SOCS-3-KO mice showed greater bone loss than WT animals, greater numbers of RANKL-positive cells, lower expression of OPG-positive cells in periodontal tissues, and greater amounts of osteoclasts [80]. The results also demonstrated that peritoneal macrophages from SOCS-3-KO mice, stimulated with lipopolysaccharide (LPS), showed higher expression of pro-inflammatory cytokines IL-1B and IL-6 in relation to macrophages from wild-type (WT) mice [80].

In contrast, JAK3 inhibition in mice with periodontal disease increased inflammatory infiltrate and bone resorption, indicating a protective role for JAK in the pathogenesis of periodontal diseases [81] (Table 3). An increase in nuclear factor kappa-light chain-enhancer of activated B cells (NF-κB) activity and the production of pro-inflammatory cytokines were also observed in the gingival tissue of animals treated with a JAK3 inhibitor [81]. Conversely, our research group demonstrated through an experimental model of periodontitis that rats treated with oral JAK inhibitors daily for seven days presented suppression of the inflammatory process and alveolar bone resorption during periodontal disease induction, suggesting that modulation of this signaling pathway may be a therapeutic approach for preventing periodontitis progression [26].

A clinical study comparing the periodontal status of patients with periodontitis and rheumatoid arthritis at baseline and 24 weeks after oral treatment with a JAK1-3 inhibitor indicated a reduction in periodontal inflammation [54]. Similarly, an improvement in periodontal clinical parameters was observed in two case reports of patients using the JAK inhibitor Tofacitinib at a dose of 10 mg/day [73]. In vitro, JAK 1 inhibitors have been shown to inhibit alkaline phosphatase activity in periodontal ligament cells stimulated by IL-11 and IL-6 with ascorbic acid [82,83].

Till now, few studies have evaluated JAK/STAT signaling pathway inhibitors in periodontal disease. However, results suggest a promising role for this pathway in regulating the pathogenesis of periodontitis, and more studies are necessary to investigate further their clinical impact in patients [85]. The adjunct use of host response modulation through the inhibition of targets, such as the JAK/STAT pathway, in conjunction with mechanical periodontal treatment may bring remarkable benefits for the treatment of periodontitis [78,85]. Figure 2 illustrates the potential use of JAK inhibitors in periodontal diseases.

### 5.4. Osteoporosis

Osteoporosis is of public health concern since it is a skeletal disorder characterized by compromised bone strength leading to an increased risk of fractures [86,87], and it affects a large part of the world’s population. The treatments currently approved for disease act largely by inhibiting bone resorption, such as using calcium and vitamin D, sex hormones, calcitonin, and bisphosphonates, or even inhibiting the activity of important mediators of bone turnover, such as receptor activator of nuclear factor-kappa B ligand (RANKL) and sclerostin, through the use of monoclonal antibodies (denosumab [88] and romosozumab [89], respectively) [90]. These therapies aim to provide multidisciplinary surveillance and symptomatic treatment of complications and sometimes resort to complicated and risky bone marrow transplant surgeries [90,91,92].

JAK inhibition has shown promising results in the field of bone health. In an initial study [21], researchers discovered that JAK inhibition increased bone mass in mice during steady-state conditions and prevented bone loss induced by estrogen deficiency (ovariectomy) in mice, by stimulating osteoblastic function. Similarly, researchers found that CYT387, a specific JAK inhibitor, inhibited osteoclast activity and alleviated ovariectomy-induced osteoporosis in mice through modulating the RANKL and ROS signaling pathways, suggesting that JAK inhibitors may be valuable in the prevention or treatment of osteoporosis. In another study [56], the JAK inhibitor, Stattic, demonstrated inhibitory effects on RANKL-mediated osteoclastogenesis in RAW 264.7 and prevented bone loss caused by ovariectomy by suppressing the activation of the STAT3 and NF-κB pathways, mediators known to play critical roles in osteoclast formation. In vitro, another important study [72] explored the role of JAK2/STAT3 in regulating estrogen-related senescence of bone marrow stem cells. The findings indicated that this signaling pathway plays a crucial role in the senescence process, suggesting that manipulation of JAK2/STAT3 could potentially influence the function of bone marrow stem cells and overall bone health. Finally, a study conducted by Fu et al. (2020) aimed to investigate the underlying mechanism of miR-151a-3p, an RNA molecule that plays a crucial role in gene expression regulation, in postmenopausal osteoporosis [72]. It demonstrated that miR-151a-3p can inhibit the expression of SOCS5, leading to the activation of the JAK2/STAT3 signaling pathway, resulting in increased osteoclastic differentiation and bone resorption. Furthermore, the research showed that overexpression of miR-151a-3p resulted in decreased bone mass and increased bone resorption [72].

Another successful case of using JAK/STAT pathway inhibitors was demonstrated in patients with Hutchinson-Gilford progeria syndrome (HGPS), an incurable condition that affects fetuses through premature aging, causing early death [93]. Progressive joint contractures, stiffness, and osteoporosis are observed, leading to a reduced life expectancy. Ex vivo experiments with human cells demonstrated that JAK inhibition restored cellular homeostasis, delayed cellular senescence, and reduced the expression levels of pro-inflammatory markers in HGPS cells [93].

Although there is limited research on the use of JAK inhibitors in the treatment of osteoporosis, these drugs have demonstrated potential in the treatment of the disease due to their ability to inhibit inflammation [94], increase osteoblast activity, and mesenchymal stromal cell differentiation, in addition to reducing osteoclast activity, which may help to prevent bone resorption [95]. However, more research is needed to gain a comprehensive understanding of the mechanisms underlying the potential benefits of JAK inhibitors in osteoporosis and to assess their efficacy and safety in patients with this condition, which affects millions of people around the world.

## 6. Conclusions

In recent years, a greater understanding of the signal transduction pathways involved in regulating cytokine production in immune cells has led to new discoveries in the development of therapies and treatments for various inflammatory diseases. One of these pathways, the JAK/STAT signaling pathway, has been the subject of numerous in vivo and in vitro studies due to its important role in the activity of several mediators relevant to the pathogenesis of osteolytic diseases.

This review highlights new discoveries about how the JAK/STAT pathway is intimately linked to the pathogenesis of a number of osteolytic diseases that affect a large part of the world’s population. In addition, it summarizes how the inhibition of this pathway has become an attractive therapeutic target for the treatment of these disorders, highlighting the progress achieved so far and creating momentum for further research on the subject.

Recent studies have provided evidence that inhibition of the JAK/STAT pathway can effectively reduce pro-inflammatory cytokine production and prevent bone destruction in several preclinical models of osteolytic diseases. These findings led to the development of several JAK/STAT inhibitors, some of which have already been approved for clinical use in the treatment of rheumatoid arthritis and other autoimmune diseases.

Clinical studies in patients with osteolytic diseases have shown that the use of these drugs significantly reduces the symptoms of the disease, although they can increase the risk of secondary infections. In animals, JAK inhibitors have been shown to reduce the expression of inflammatory mediators, improving the signs and symptoms of inflammatory diseases such as rheumatoid arthritis and myelofibrosis. In vitro research also suggests that these drugs may reduce inflammation in bone cells and decrease the formation of bone-destroying cells, which may hold promise for treating these diseases. In periodontitis, although data from preclinical animal studies are conflicting, clinical studies have indicated improvement in periodontal parameters in patients using JAK inhibitors. Overall, these studies demonstrate the potential to target the JAK/STAT pathway as a promising therapeutic approach. However, it is important to conduct randomized controlled clinical trials to assess the safety and efficacy of these drugs, in humans, in addition to analyzing important aspects such as the duration of treatment, proper dosage, and possible side effects of these drugs in order to indicate protocols that are most suitable and efficient for the clinical use of JAK inhibitors in osteolytic diseases.

## Figures and Tables

**Figure 1 ijms-24-10290-f001:**
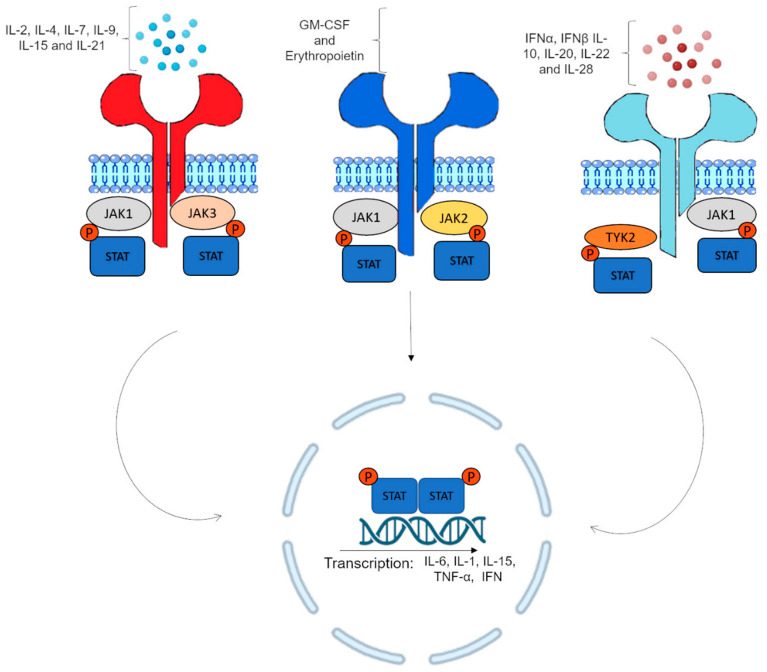
Potential mechanisms for activation of the JAK-STAT signaling pathway. Once the cytokines bind to their receptors with intracellular signaling via Janus kinase, it triggers an intracellular signal transducer that causes the autophosphorylation and phosphorylation of the cytoplasmic domain of cytokine receptors associated with JAKs (JAK1, JAK2, JAK3, and TYK2) and also triggers the phosphorylation of the STAT proteins. JAK-mediated STAT phosphorylation results in the phosphorylation and dimerization of STAT protein, nuclear translocation, and induction of transcription of inflammatory cytokines such as interleukins (IL)-1, IL-6, IL-15, tumor necrosis factor alpha (TNF-α), and interferons (IFN).

**Figure 2 ijms-24-10290-f002:**
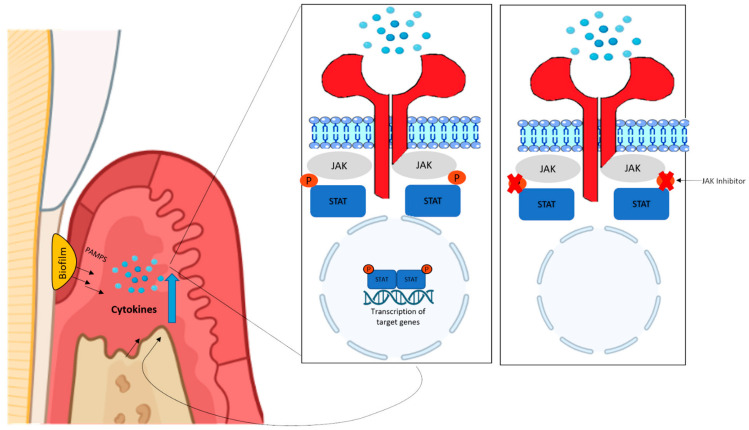
The recognition of microbial components (molecular patterns associated with pathogens—PAMPs) by defense cells induces an increase in inflammatory cytokines in periodontal tissues. These cytokines stimulate the activation of the JAK-STAT pathway, which results in the transduction of more inflammatory cytokines, amplifying the inflammatory response and tissue destruction. JAK inhibitors bind to and competitively inhibit the kinase domain of JAKs, thereby preventing JAKs from phosphorylating STATs and other substrates so that intracellular signals cannot be further transduced. Because JAKs are critical for multiple different cytokines, JAK inhibitors can block the action of a range of cytokines and contribute to the reduction of the inflammatory response and tissue destruction in the periodontium.

**Table 3 ijms-24-10290-t003:** Summarization of the findings from periodontal disease studies.

Drugs	Study Model	Outcomes
**JAK3 Inhibitor** [26]	**in vivo**	Suppression of the inflammatory process and alveolar bone resorption during periodontal disease induction;
**JAK1 inhibitor** [82]	**in vitro**	Alkaline phosphatase activity inhibition in IL-11/ascorbic acid stimulated periodontal ligament cells;
**AG490 and JAK1 inhibitor** [83]	**in vitro**	Both AG490 and JAK inhibitor I significantly diminished ascorbic acid+IL-6/sIL-6R-elicited alkaline phosphatase activity;
**JAK1 and STAT3 inhibitors** [84]	**in vitro**	No observed effect
**JAK 3 inhibitor** [81]	**in vivo** and **in vitro**	**in vivo**: - Enhanced infiltration of inflammatory cells, reduced expression of Wnt3a and Dvl3 in *P. gingivalis*-infected gingival tissues, and increased disease severity; **in vitro**: - Enhancement of nuclear factor kappa-light-chain-enhancer of activated B cells activity and the production of pro-inflammatory cytokines (TNFα, IL-6 e IL-12P40) in *P. gingivalis*-stimulated innate immune cells.

Abbreviations: IL: Interleukins; TNFα: Tumor Necrosis Factor alpha.

## Data Availability

Data sharing is not applicable.

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
