# Peer review of "JAK/STAT as a Potential Therapeutic Target for Osteolytic Diseases"

_ijms, 2023, doi:10.3390/ijms241210290_

Round 1

Reviewer 1 Report

"This study presents the importance of JAK/STAT signaling pathway in inflammation-induced bone resorption, the results of relevant clinical studies, and experimental models of JAK inhibitors in osteolytic diseases. Authors emphasized the bone metabolism through JAK/STAT signaling pathway, and introduced the inhibitors used in specific diseases such as rheumatoid arthritis, myelofibrosis, periodontal disease, and osteoporosis.

This is an interesting work, which can be of help for understanding of JAK/STAT pathway and its association with osteolytic diseases. Nevertheless, results from all existing studies are too preliminary to make a review article. The manuscript is well written, but there were some issues to be addressed. The conclusion section cannot properly summarize this review. It would be better to write the section more comprehensively and to propose any future study.

Line 335-375: Please add the details of clinical studies, such as medication doses.

Line 501: I’d like to suggest to add other medications for osteoporosis, such as denosumab, and romosozumab.

Line 527-537: The conclusions related to each disease are written in each section, but it would be better to summarize and emphasize the conclusions."

Author Response

Dear Reviewer,

We appreciate the time and consideration dedicated to read our manuscript and kindly invite you to consider our point-by-point response to your queries and comments below.

All changes in the revised manuscript are highlighted in yellow.

Reviewer #1:

  1. Comment: “Nevertheless, results from all existing studies are too preliminary to make a review article.”

Our reply/ answer:  We agree with the reviewer that many studies on the JAK/STAT pathway in osteolytic diseases are still preliminary, and that certainly important information on this subject will be able to add relevant knowledge in the near future, with the performance of a larger number of studies, however, considering the numerous JAK inhibitors currently approved and the increase in their clinical use, it is crucial to clarify the importance of the pathway, its mechanism of action and its relevance on the pathogenesis of inflammatory bone diseases, for understanding the role of inhibitors as a therapeutic strategy , and the results obtained so far. This knowledge may guide further preclinical and clinical research on the use of JAK inhibitors, and help to consolidate the knowledge obtained so far.

  1. Comment: “The conclusion section cannot properly summarize this review. It would be better to write the section more comprehensively and to propose any future study.”

Our reply/ answer: As recommended, the conclusion section is now more comprehensive and clearer, with more specific proposals for future studies.

  1. Comment: “Line 335-375: Please add the details of clinical studies, such as medication doses.”

Our reply/ answer: As recommended, details from clinical studies (such as the dose of drugs used) have been added.

  1. Comment: “Line 501: I’d like to suggest to add other medications for osteoporosis, such as denosumab, and romosozumab.”

Our reply/ answer: As recommended, suggested medications were added to the session.

  1. Comment: “Line 527-537: The conclusions related to each disease are written in each section, but it would be better to summarize and emphasize the conclusions."

Our reply/ answer: As recommended, the conclusions are now better summarized and emphasized.

Reviewer 2 Report

The article is trying to review the new findings around the importance of JAK/STAT signaling in disease related bone loss. 

I have some suggestions for impoving the article:

In line 51 instead of "during bone inflamatory remodeling" I would say bone remoddeling during inflamation or bone remodelling affected by inflamation. 

In figure 1 caption what is the meaning of transduction of inflamatory cytokines? do you mean induction of transcription of inflamatory cytokines?

Line 145-149 is not very clear paragraph. It would be great to explain the effects and mechanism(s).

In section 3 ( bone metabolism) I suggest to add more studies that induction of JAK/STAT trigger more bone resorption. 

lines 509-512, the paragraph is vague. you may need to explain what mechanism is involved or what experiments have been done to show what? 

I suggest to update your review article with more recent studies that have been done in 2021-2023. 

Best wishes!

The quality of English language is good but it needs to be more scientific.  

Author Response

Dear Reviewer,

We appreciate the time and consideration dedicated to read our manuscript and kindly invite you to consider our point-by-point response to your queries and comments below.

All changes in the revised manuscript are highlighted in yellow.

Reviewer #2:

  1. Comment: “In line 51 instead of "during bone inflamatory remodeling" I would say bone remoddeling during inflamation or bone remodelling affected by inflammation.”

Our reply/ answer: As suggested, we changed the expression used for better understanding.

  1. Comment: “In figure 1 caption what is the meaning of transduction of inflamatory cytokines? do you mean induction of transcription of inflamatory cytokines?”

Our reply/ answer: The expression 'transduction of inflammatory cytokines' in the legend of figure 1 was used to describe the process of activation of the JAK-STAT signaling pathway in response to the binding of cytokines to their receptors, and induction of transcription of inflammatory cytokines. We have modified the passage to 'transcription of inflammatory cytokines' to make it clearer to the reader. We appreciate the comment.

  1. Comment: “Line 145-149 is not very clear paragraph. It would be great to explain the effects and mechanism(s).”

Our reply/ answer: As recommended, the paragraph is now better highlighting the effects and mechanisms mentioned.

  1. Comment: “In section 3 (bone metabolism) I suggest to add more studies that induction of JAK/STAT trigger more bone resorption.”

Our reply/ answer: As recommended, more studies have been added in the bone metabolism section.

  1. Comment: “lines 509-512, the paragraph is vague. you may need to explain what mechanism is involved or what experiments have been done to show what?”

Our reply/ answer: As recommended, with more articles on the subject the paragraph is now clearer, highlighting the mechanisms of the study as well as its purpose.

  1. Comment: “I suggest to update your review article with more recent studies that have been done in 2021-2023.”

Our reply/ answer: We are grateful for your valuable suggestion regarding updating the papers used in the review. We fully agree that it is crucial to ensure that the information presented is up-to-date and relevant to the scientific community. Carefully reviewing the literature for recent articles related to osteolytic diseases and JAK/STAT pathway inhibitors, however, unfortunately, we found few articles published in the last two years. However, when using the keywords and filters that you can check below, we found some more articles related to arthritis, myelofibrosis and osteoporosis, which we believe are relevant to this review. We hope that, with these additions, the article will be completer and more up-to-date.

Keywords:

(JAK) AND (inhibition) AND (osteolytic disease)

(STAT) AND (inhibition) AND (osteolytic disease)

(((JAK) OR (STAT)) AND (inhibition)) AND (osteolytic disease)

(((JAK) OR (STAT)) AND (osteolytic disease)

(JAK) AND (osteolytic disease)

(STAT) AND (osteolytic disease)

(JAK) AND (inhibition) AND (arthritis)

(STAT) AND (inhibition) AND (arthritis)

(((JAK) OR (STAT)) AND (inhibition)) AND (arthritis)

(((JAK) OR (STAT)) AND (arthritis)

(JAK) AND (arthritis)

(STAT) AND (arthritis)

(JAK) AND (inhibition) AND (myelofibrosis)

(STAT) AND (inhibition) AND (myelofibrosis)

(((JAK) OR (STAT)) AND (inhibition)) AND (myelofibrosis)

(((JAK) OR (STAT)) AND (myelofibrosis)

(JAK) AND (myelofibrosis)

(STAT) AND (myelofibrosis)

(JAK) AND (inhibition) AND (periodontitis)

(STAT) AND (inhibition) AND (periodontitis)

(((JAK) OR (STAT)) AND (inhibition)) AND (periodontitis)

(((JAK) OR (STAT)) AND (periodontitis)

(JAK) AND (periodontitis)

(STAT) AND (periodontitis)

(JAK) AND (inhibition) AND (periodontal disease)

(STAT) AND (inhibition) AND (periodontal disease)

(((JAK) OR (STAT)) AND (inhibition)) AND (periodontal disease)

(((JAK) OR (STAT)) AND (periodontal disease)

(JAK) AND (periodontal disease)

(STAT) AND (periodontal disease)

(JAK) AND (inhibition) AND (osteoporosis)

(STAT) AND (inhibition) AND (osteoporosis)

(((JAK) OR (STAT)) AND (inhibition)) AND (osteoporosis)

(((JAK) OR (STAT)) AND (osteoporosis)

(JAK) AND (osteoporosis)

(STAT) AND (osteoporosis)

Filters applied: Clinical Trial, Randomized Controlled Trial.

Papers published between 2021-2023.

Reviewer 3 Report

Review of “JAK/STAT as a Potential Therapeutic Target for Osteolytic Diseases” by Mariely A. Godoi et al 2023

Authors present a review about JAK/STAT as a Potential Therapeutic Target for Osteolytic Diseases well structured adding information about therapeutic approaches of JAK/STAT signal pathway, however manuscript still have some issues to be addressed:

Abbreviations of genes and proteins should be reviewed, the description of full name of some are missing like TNK-alpha at first mention, present only in table 3 after several mentions.

Authors mention that “Other studies have demonstrated the role of the same pathway in osteoblasts formation due to its strong performance in important transcription factors in this process, such as Runx-2, BMP-7, and Tbx-3 [12,18–20] “ If authors are referring to proteins, abbreviations should be corrected, all capital letters, if refers to genes, must be in italic accordingly with species rules, by the text, seems proteins references, so….In addition, full name for the first time is missing.

Tables are not well included in the manuscript, probably as consequence of manuscript pdf production.

Authors mention that “When evaluating SOCS3 knockout mice, increased alveolar bone loss due to a greater number of osteoclasts and increased expression of RANKL/OPG during the progression of periodontal disease have been demonstrated “, which suggest the wrong idea, actually the balance between the expression of RANKL and OPG can be a marker for bone turnover, in the case of the study described by the authors RANKL expression increases and OPG expression decreases, which favors bone resorption, in means that saying that “increased expression of RANKL/OPG during the progression of periodontal disease” is not well explained, should be referred that the ration between the expression of RANKL and OPG favors RANKL.

Authors mention that “ Two other important bone diseases that affect a large part of the world population are osteoporosis and osteopetrosis”, however never mention if any type of therapeutic approach was done through JAK/STAT pathway for inhibition of bone formation in the case of osteopetrosis, and should mention applications for osteoporosis. If there are no application yet authors should mention this or add any application involving JAK/STAT pathway to this section.

Minor editing of English language required

Author Response

Dear Reviewer,

We appreciate the time and consideration dedicated to read our manuscript and kindly invite you to consider our point-by-point response to your queries and comments below.

All changes in the revised manuscript are highlighted in yellow.

Reviewer #3:

  1. Comment: “Abbreviations of genes and proteins should be reviewed, the description of full name of some are missing like TNK-alpha at first mention, present only in table 3 after several mentions.”

Our reply/ answer: As recommended, descriptions of full name of genes were added to the text the first time they were mentioned.

  1. Comment: “Authors mention that “Other studies have demonstrated the role of the same pathway in osteoblasts formation due to its strong performance in important transcription factors in this process, such as Runx-2, BMP-7, and Tbx-3 [12,18–20] “ If authors are referring to proteins, abbreviations should be corrected, all capital letters, if refers to genes, must be in italic accordingly with species rules, by the text, seems proteins references, so….In addition, full name for the first time is missing.”

Our reply/ answer: As recommended, because abbreviations refer to proteins, we put them all in capital letters.

  1. Comment: “Tables are not well included in the manuscript, probably as consequence of manuscript pdf production.”

Our reply/ answer: The table is now better included in the manuscript (Word).

  1. Comment: “Authors mention that “When evaluating SOCS3 knockout mice, increased alveolar bone loss due to a greater number of osteoclasts and increased expression of RANKL/OPG during the progression of periodontal disease have been demonstrated “, which suggest the wrong idea, actually the balance between the expression of RANKL and OPG can be a marker for bone turnover, in the case of the study described by the authors RANKL expression increases and OPG expression decreases, which favors bone resorption, in means that saying that “increased expression of RANKL/OPG during the progression of periodontal disease” is not well explained, should be referred that the ration between the expression of RANKL and OPG favors RANKL.”

Our reply/ answer: The paragraph in which we cite the study in question was rewritten with the aim of making the outcomes clearer and highlighting the most relevant data.

  1. Comment: “Authors mention that “Two other important bone diseases that affect a large part of the world population are osteoporosis and osteopetrosis”, however never mention if any type of therapeutic approach was done through JAK/STAT pathway for inhibition of bone formation in the case of osteopetrosis, and should mention applications for osteoporosis. If there are no application yet authors should mention this or add any application involving JAK/STAT pathway to this section.”

Our reply/ answer: Upon carefully reviewing the literature, we did not find any studies linking the JAK/STAT signaling pathway and the disease osteopetrosis, so we removed it from the sentence and added a paragraph mentioning the applications of the JAK/STAT pathway incursion in osteoporosis.

Round 2

Reviewer 1 Report

This manuscript was revised according to the reviewers' comments and its quality was improved.

Author Response

Reviewer 1# We thank the reviewer for the time dedicated to reading our manuscript and suggestions to improve its quality.